# A 3D adult zebrafish brain atlas (AZBA) for the digital age

Justin W Kenney[1,2]*[†], Patrick E Steadman[2†], Olivia Young[1], Meng Ting Shi[1], Maris Polanco[1], Saba Dubaishi[1], Kristopher Covert[1], Thomas Mueller[3], Paul W Frankland[2,4,5,6]*

[1]Department of Biological Sciences, Wayne State University, Detroit, United States; [2]Program in Neurosciences and Mental Health, The Hospital for Sick Children, Toronto, Canada; [3]Division of Biology, Kansas State University, Manhattan, United States; [4]Department of Physiology, University of Toronto, Toronto, Canada; [5]Institute of Medical Sciences, University of Toronto, Toronto, Canada; [6]Department of Psychology, University of Toronto, Toronto, Canada

**Abstract** Zebrafish have made significant contributions to our understanding of the vertebrate brain and the neural basis of behavior, earning a place as one of the most widely used model organisms in neuroscience. Their appeal arises from the marriage of low cost, early life transparency, and ease of genetic manipulation with a behavioral repertoire that becomes more sophisticated as animals transition from larvae to adults. To further enhance the use of adult zebrafish, we created the first fully segmented three-dimensional digital adult zebrafish brain atlas (AZBA). AZBA was built by combining tissue clearing, light-sheet fluorescence microscopy, and three-dimensional image registration of nuclear and antibody stains. These images were used to guide segmentation of the atlas into over 200 neuroanatomical regions comprising the entirety of the adult zebrafish brain. As an open source, online (azba.wayne.edu), updatable digital resource, AZBA will significantly enhance the use of adult zebrafish in furthering our understanding of vertebrate brain function in both health and disease.

**\*For correspondence:**
jkenney9@wayne.edu (JWK);
paul.frankland@sickkids.ca (PWF)

[†]These authors contributed equally to this work

**Competing interest:** The authors declare that no competing interests exist.

## Editor's evaluation

The issue of building a high-quality brain atlas for vertebrates has been a long-standing challenge in the field. Your work has nicely hit this mark using zebrafish, and using a method that should be applicable to many different stages and other organisms.

## Introduction

Uncovering general principles of neuroanatomical function and brain-behavior relationships requires the integration of findings across model organisms that range in complexity, organization, and accessibility (*Brenowitz and Zakon, 2015*; *Marder, 2002*; *Yartsev, 2017*). Amongst vertebrate model organisms in neuroscience, zebrafish are relative newcomers that have grown in popularity in recent years (*Kenney, 2020*; *Orger and de Polavieja, 2017*). Originally established as a model organism for developmental biology due to ease of domestication, high fecundity, and early life transparency (*Parichy, 2015*), the increased popularity of zebrafish is driven by recent advancements in brain imaging, molecular genetic manipulation, and behavior. To further enhance the use of zebrafish as an animal model in neuroscience, we created a digital three-dimensional (3D) brain atlas (adult zebrafish brain atlas [AZBA]). Although several digital atlases exist for larval zebrafish (*Kunst et al., 2019*; *Randlett et al., 2015*; *Ronneberger et al., 2012*; *Tabor et al., 2019*), no atlas of equivalent detail has been created

for adult zebrafish. Adult zebrafish provide several important advantages over larval zebrafish such as a mature and histologically differentiated neuroanatomy and a rich behavioral repertoire that includes long-term associative memory, complex social interactions, and goal-driven behaviors (*Gerlai, 2016*; *Kalueff et al., 2013*; *Kenney et al., 2017*; *Nakajo et al., 2020*).

3D digital brain atlases are essential tools for modern neuroscience because they facilitate lines of inquiry that are not possible with two-dimensional book-based atlases. For example, visualization of the 3D structure of the brain and incorporation of new data or discoveries are difficult, if not impossible, with a print atlas. In contrast, digital atlases enable exploration of brain structures in any arbitrary 3D perspective and can be readily updated to incorporate new information such as patterns of gene expression and anatomical connectivity, as has been done for the mouse (*Wang et al., 2020*). Such features are important for fields like neurodevelopment and comparative neuro-anatomy that rely on 3D topologies to understand how specific brain regions develop and relate across species. A digital atlas would also enhance the use of adult zebrafish in disease modeling by enabling a more comprehensive understanding of how the brain changes in response to insults that occur in neurodevelopmental disorders (*Sakai et al., 2018*), traumatic brain injury (*McCutcheon et al., 2017*), neurodegeneration (*Xi et al., 2011*), and the formation and spreading of gliomas (*Idilli et al., 2017*). Finally, digital atlases enable automated segmentation of new images, which is essential for whole brain mapping approaches that can lead to unexpected discoveries of regional function (*Bahl and Engert, 2020*; *Kim et al., 2015*; *Pantoja et al., 2020*; *Randlett et al., 2019*). Whole brain activity mapping also facilitates powerful network approaches to understanding brain (*Bassett and Sporns, 2017*; *Coelho et al., 2018*; *Vetere et al., 2017*; *Wheeler et al., 2013*), yielding insight into how coordinated brain-wide activity gives rise to behavior.

The generation of a brain atlas for adult zebrafish presents several challenges in comparison to the larval brain because the mature brain is opaque and an order of magnitude larger; this makes traditional whole-mount approaches impossible and confocal or wide-field microscopy infeasible. We overcome these roadblocks by exploiting recent developments in histology and microscopy. To enable whole-mount imaging, we rendered adult brains transparent using a tissue clearing approach (iDISCO+) that is compatible with different stains such as small molecules, anti-bodies, and in situ probes (*Kramer et al., 2018*; *Renier et al., 2016*). High-resolution imaging of samples as large as intact adult brains is not amenable to conventional microscopic techniques,

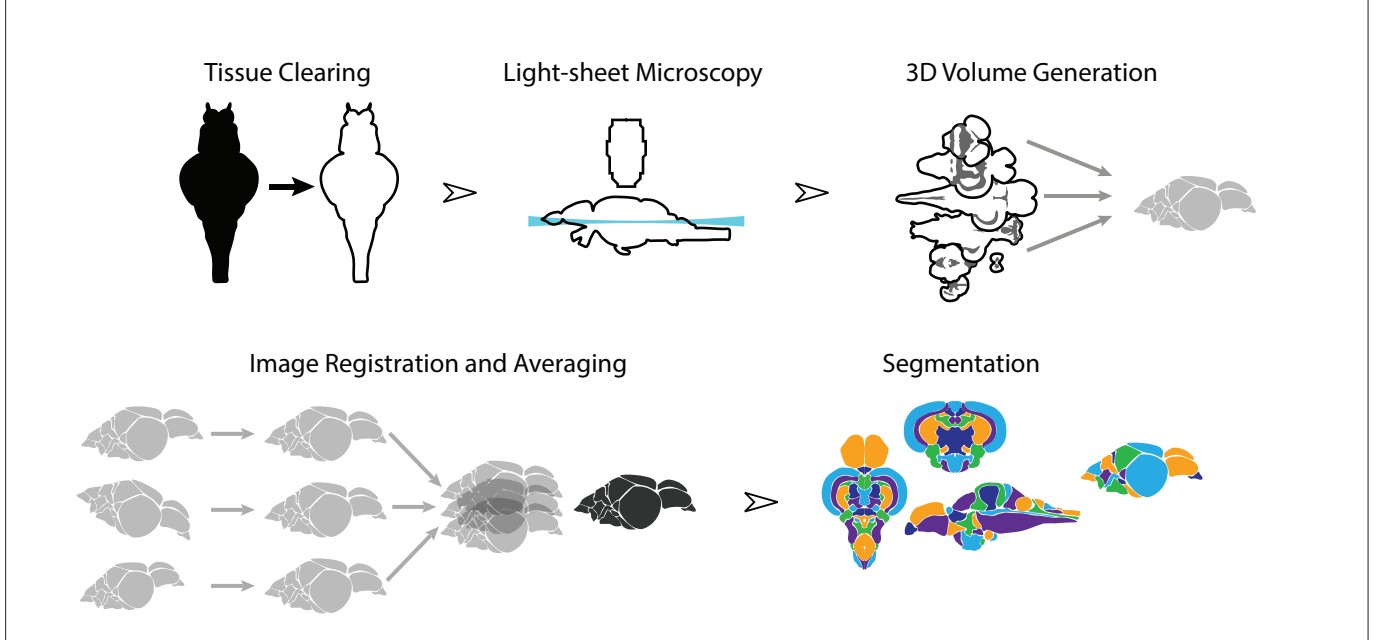

**Figure 1.** Overview of the strategy for generating the adult zebrafish brain atlas (AZBA). Dissected brain samples were first subject to staining and tissue clearing. This was followed by whole-mount imaging using light-sheet fluorescence microscopy. Three-dimensional volumes were created from individual image sets, and then registered into the same anatomical space prior to averaging to generate a representative image. Finally, volumes were segmented into over 200 neuroanatomical regions.

so we turned to light-sheet fluorescence microscopy for rapid large volume imaging with minimal photobleaching (*Pitrone et al., 2013*; *Reynaud et al., 2014*). To generate an atlas that minimizes individual variations in neuroanatomy, we registered 3D volumes from multiple animals into the same anatomical space prior to averaging (*Lerch et al., 2011*). To help guide segmentation and generate insight into the neurochemical organization of the brain, images from 10 different antibody stains were also registered into the same anatomical space. Finally, we performed manual segmentation, delineating the atlas into over 200 neuroanatomical regions, including nuclei and white matter tracts.

Taken together, AZBA is the most comprehensive, detailed, and up-to-date atlas of the adult zebrafish brain. We have made all averaged images freely available (https://doi.org/10.5061/dryad.dfn2z351g; azba.wayne.edu) to enable their use in exploring the organization of the zebrafish brain and automated segmentation for activity mapping. By generating this resource using readily available techniques, AZBA can be continuously updated to reflect the latest findings in zebrafish neuroanatomy. We anticipate this becoming an indispensable resource as adult zebrafish continue to gain traction as a model organism in understanding the intricacies of the vertebrate brain.

## Results

### Overall strategy

To create an averaged 3D atlas, we developed a sample preparation and analysis pipeline for whole-mount 3D image acquisition and registration (*Figure 1*). We used a whole-mount preparation to avoid issues with slice-based techniques such as tissue loss, tearing, and distortion. To circumvent the challenge of tissue opacity, we used a rapid organic solvent-based tissue clearing technique, iDISCO+ (*Renier et al., 2016*), that renders brains optically transparent. Because conventional microscopic techniques are not suitable for large volume imaging, we used light-sheet microscopy. Image stacks from individual fish brains were converted to 3D volumes and registered into the same anatomical space prior to averaging. Finally, average 3D images were manually segmented into their constituent brain regions.

### Light-sheet imaging

Tissue clearing using iDISCO+ resulted in transparent brains (*Figure 2A*). iDISCO+ is compatible with a variety of stains, such as a nuclear stain (TO-PRO), that allowed us to approximate the Nissl

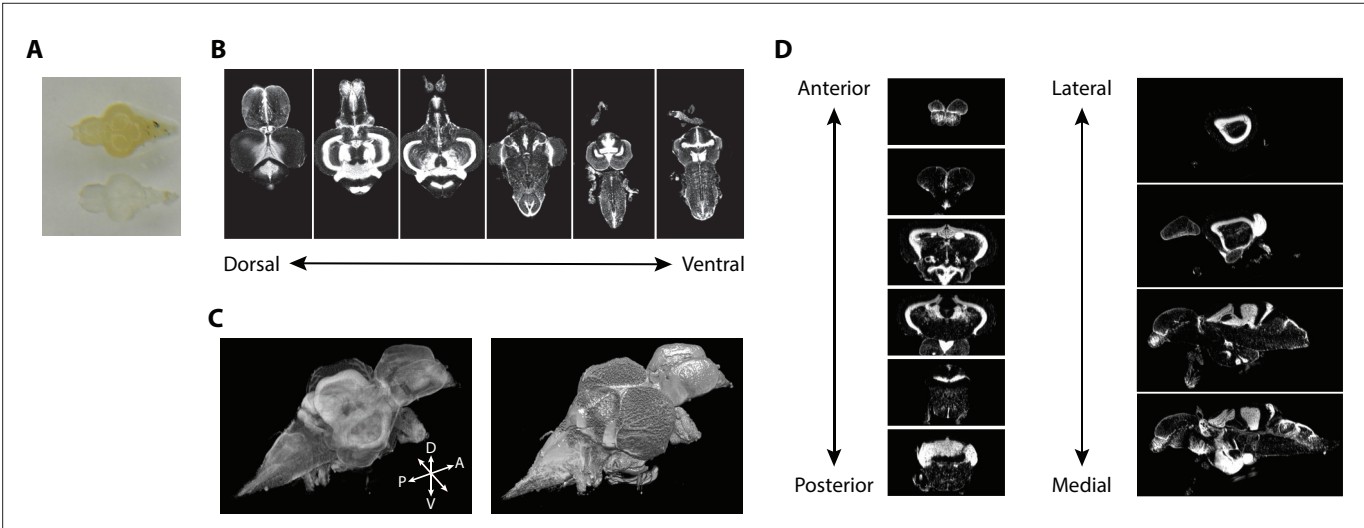

**Figure 2.** Imaging of nuclear-stained tissue-cleared samples. (**A**) Image of adult zebrafish brain samples before (top) and after (bottom) clearing using iDISCO+. (**B**) Example TO-PRO-stained images from a single sample acquired in the horizontal plane during light-sheet imaging. (**C**) Three-dimensional volumes generated from a set of light-sheet images from an individual brain visualized using a maximum intensity projection (left) and exterior volume (right). (**D**) Coronal (left) and sagittal (right) views of an individual brain generated from a single three-dimensional volume.

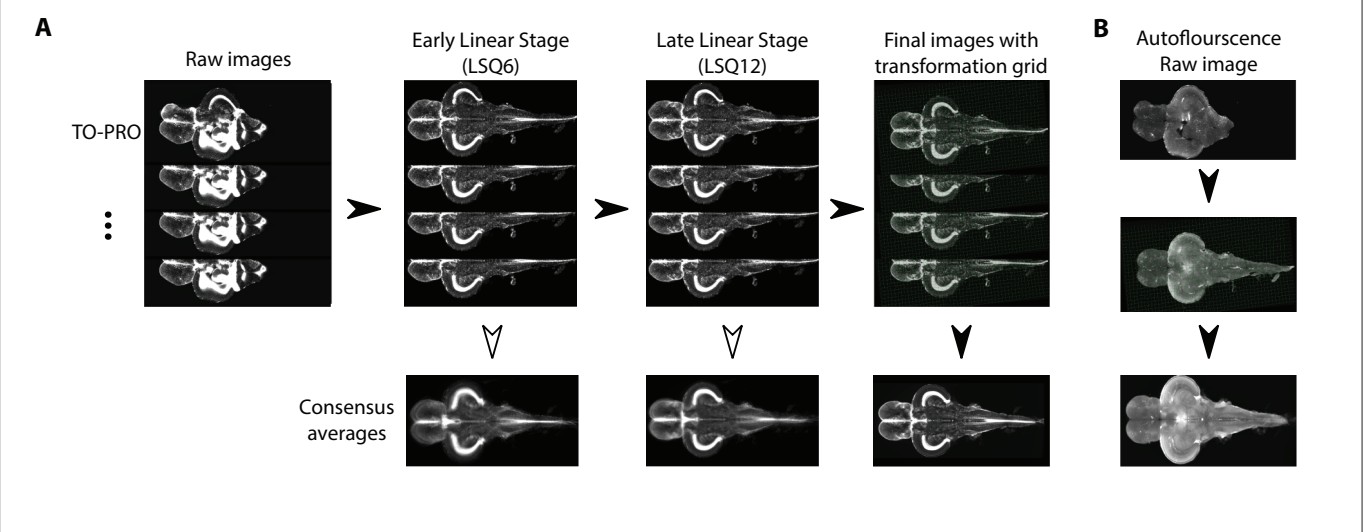

**Figure 3.** Image registration pipeline. (**A**) Raw TO-PRO images from 17 fish were first aligned using linear transformations (LSQ6 and LSQ12) followed by a final nonlinear transformation (right). Deformation grids at each stage are overlaid. Consensus average images at each stage of the pipeline are given below. (**B**) Raw autofluorescence images (top) acquired at the same time as the TO-PRO images were registered into the same space using the transformations derived from TO-PRO registration (middle). Images were then averaged together to generate a corresponding autofluorescence average in the same anatomical space as the TO-PRO images (bottom).

stain in the print atlas (*Wullimann et al., 1996*). Cleared and stained brains were imaged in the horizontal plane with an in-plane resolution of 3.25 μm and an axial step size of 4 μm (*Figure 2B*) yielding near-isotropic signals at sufficient resolution to clearly distinguish regional boundaries. From this collection of images, we generated 3D volumes (*Figure 2C*) that enabled viewing at any arbitrary angle including the coronal and sagittal planes (*Figure 2D*). Images were subject to quality control so that those damaged during dissection were discarded. We retained 17 sets of nuclear-stained and associated autofluorescence images from both male and female fish (eight females), each of which were transformed into 3D volumes for registration.

## 3D image registration

The atlas was generated by registering images from individual animals into the same space, thereby creating an anatomical average. This approach has been previously used in humans (*Klein et al., 2009*), mice (*Dorr et al., 2008*; *Steadman et al., 2014*), macaques (*McLaren et al., 2009*), and zebrafish larvae (*Kunst et al., 2019*; *Randlett et al., 2015*; *Ronneberger et al., 2012*; *Tabor et al., 2019*). We used the contrast from the TO-PRO signal and an image registration pipeline toolkit (*Friedel et al., 2014*) to perform iterative registration to generate a consensus image (*Figure 3*). This method begins with a 6-parameter linear registration to rotate and translate the initial image dataset followed by a 12-parameter affine registration to scale, translate, rotate, and shear the dataset with a pairwise approach to avoid bias by outlier images (*Figure 3A*). Lastly, an iterative nonlinear registration with six iterations at subsequently higher resolutions was performed using minctracc (*Collins and Evans, 2011*). This resulted in a set of linear and nonlinear transformations for each TO-PRO image in our dataset from native space to a consensus space and orientation. These transformations were then applied to corresponding autofluorescence images, thereby creating an atlas with averaged images containing TO-PRO and autofluorescence signals (*Figure 3B*).

## Antibody stains

To provide additional guidance for segmentation and generate insight into the neurochemical organization of the adult zebrafish brain, we also acquired images using 10 different antibody stains (*Figure 4A*). We sought stains that would identify different cell types in the brain, such as neurons (HuC/D), radial glial cells (glial fibrillary associated protein [GFAP]), and proliferating cells (proliferating cell nuclear antigen [PCNA]), markers for different neurotransmitters (tyrosine hydroxylase [TH], 5-hydroxytryptamine [5-HT], and choline acetyltransferase [ChAT]), and calcium

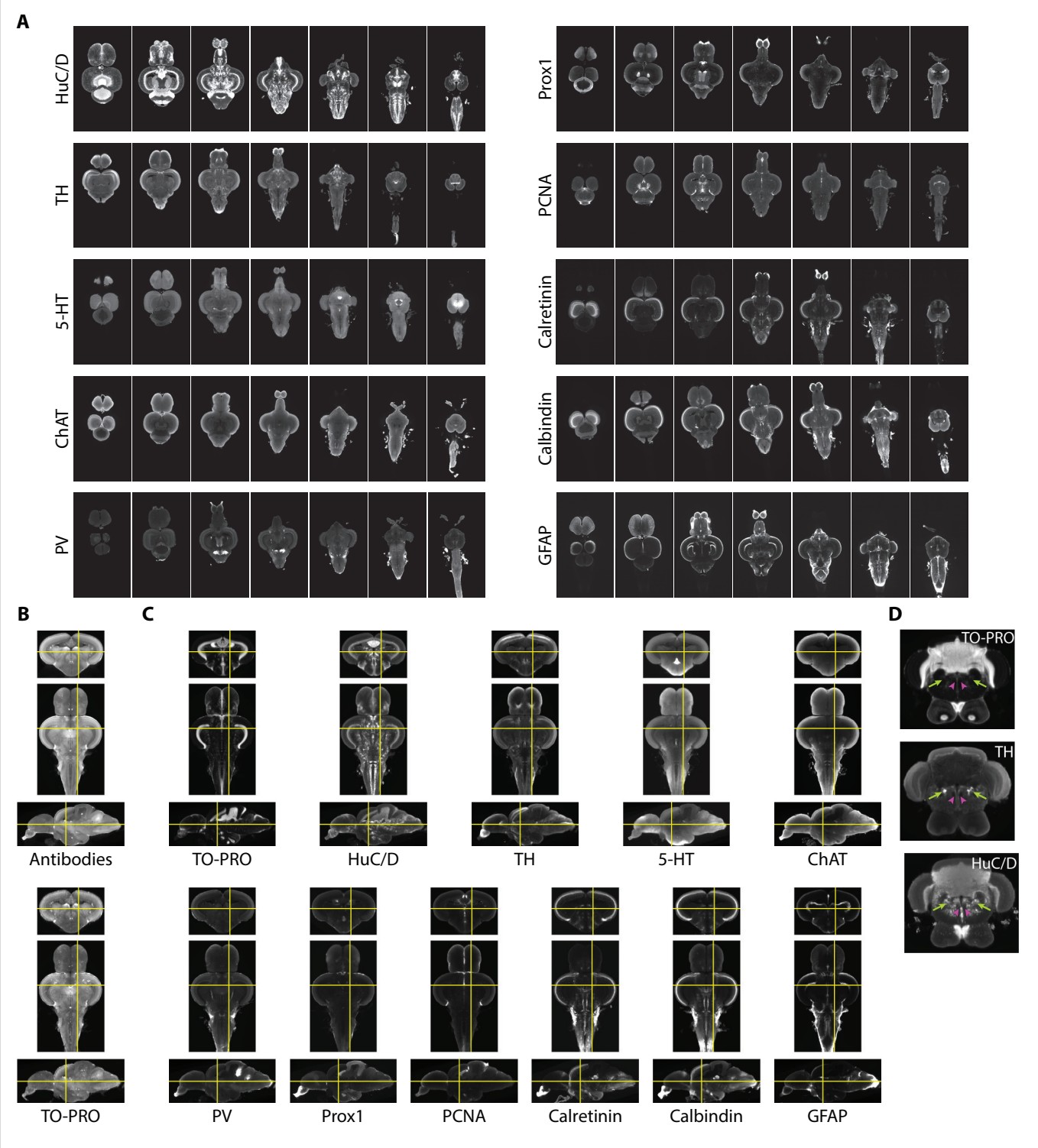

**Figure 4.** Imaging and registration of antibody stains. (**A**) Representative light-sheet images taken in the horizontal plane from individual brains stained with indicated antibodies. (**B**) Autofluorescence images acquired during antibody staining (top) were registered into the same space as autofluorescence images acquired during TO-PRO staining (bottom). (**C**) Transformations from autofluorescence registration were applied to antibody images to bring antibody stains into the same anatomical space as the TO-PRO stain. Yellow crosshairs are in the same place on each image. (**D**) Example of correspondence between TO-PRO and antibody images and how stains can be used to identify the boundaries of specific nuclei (green arrow: locus coeruleus) and white matter tracts by a lack of staining (pink arrowhead: medial longitudinal fascicle).

The online version of this article includes the following figure supplement(s) for figure 4:

*Figure 4 continued on next page*

*Figure 4 continued*
**Figure supplement 1.** Registration precision between six different landmarks in the autofluorescence and TO-PRO images and their respective atlas images.

binding proteins (parvalbumin [PV], calbindin, and calretinin). Some of these stains, such as TH, 5-HT, ChAT, and calretinin, have already been subject to brain-wide analysis, making them useful for guiding segmentation.

Fully realizing the utility of different stains requires images to be brought into the same anatomical space as the previously generated TO-PRO average. To achieve this, during imaging of antibody stains, we also acquired autofluorescence images, thereby providing a bridge between the antibody images and the TO-PRO images. Autofluorescence images from antibody stains were registered with the autofluorescence channel of the TO-PRO images, yielding a set of transformations that were used to bring antibody stains into the same anatomical space as the TO-PRO stain (*Figure 4B*). To generate a representative image for each antibody, we averaged together at least five independent brains. Our approach resulted in strong correspondence between antibody images and the TO-PRO stain (*Figure 4C*). The utility of this approach is apparent from examining structures known to express high levels of specific enzymes, like TH in the locus coeruleus (*Figure 4D*; green arrows).

## Registration precision

To compare registration precision using TO-PRO and autofluorescence signals, we labeled six different landmarks in the atlas images and corresponding points in acquired images. Transforms derived from the registration process were then applied to the acquired image's labeled landmarks. We then measured the Euclidean distance between the transformed points and the points in the atlas. Using a mixed-model ANOVA (2 × 15; signal [between subject] × landmark [within subject]), we found a main effect of signal ($F(1,13) = 1084$, $p=6.6 \times 10^{-14}$) with the TO-PRO signal having greater mean precision (15 ± 10 µm vs. 99 ± 53 µm; *Figure 4—figure supplement 1*). However, there was also a significant effect of landmark ($F(5,65) = 98$, $p<2 \times 10^{-16}$), and an interaction between signal and landmark ($F(5, 65) = 120$, $p<2 \times 10^{-16}$). A closer examination of the data revealed that in the autofluorescence images the average precision of each landmark covers a much wider range (17–180 µm) than TO-PRO (9–23 µm). In the autofluorescence image, the landmark with the highest precision (point 5; 17 ± 4 µm) has precision on par with the TO-PRO average. This suggests that the larger error measured using autofluorescence images is likely due to experimenter error in selecting points, reflecting the paucity of well-defined landmarks in this signal compared to the richer, high-contrast TO-PRO images.

## Segmentation

Registered images were used to segment the brain into its constituent parts (*Figure 5*; see *Supplementary file 1* for anatomical abbreviations and colors). Segmentation was primarily guided by the seminal atlas of *Wullimann et al., 1996*. Regional boundaries and terminology were updated for parts of the brain that have been subject to more recent analysis such as the telencephalon, hypothalamic regions, and motor nuclei (*Mueller et al., 2004*; *Porter and Mueller, 2020*; *Rink and Wullimann, 2001*). Segmenting large, clearly delineated regions, such as the optic tectum and the cerebellum, was straightforward (*Figure 5A*). Small nuclei that only appear in one or two slices in the atlas or in images from only one axis proved more challenging. For such regions, we primarily made use of the coronal axis due to it being the most extensively represented in both the atlas and the literature (*Figure 5A*, bottom). The horizontal and sagittal planes enabled us to identify the anterior-posterior boundaries (*Figure 5A*, top and middle). To ensure we captured as many neuronal structures as possible, we also made extensive use of a neuronal marker (HuC/D) in conjunction with the nuclear stain, which allowed us to safely identify many boundaries (*Figure 5—figure supplement 1*). Other challenges included the fact that the original brain atlas contains a significant amount of unsegmented space. We labeled these regions as 'unknown' and according to their lowest anatomical level (e.g., unknown ventral telencephalon [UnkVT], unknown diencephalon [UnkD], etc.). Explicitly labeling these regions distinguishes them from the 'clear' label that is used for areas outside the brain to facilitate computational analysis using the atlas. Identification of tracts was largely based on a combination of autofluorescence and lack of nuclear and neuronal staining since we were unsuccessful in finding a white matter stain compatible with iDISCO+ (e.g., the MLF: *Figure 4D*, pink arrowheads). We anticipate that future

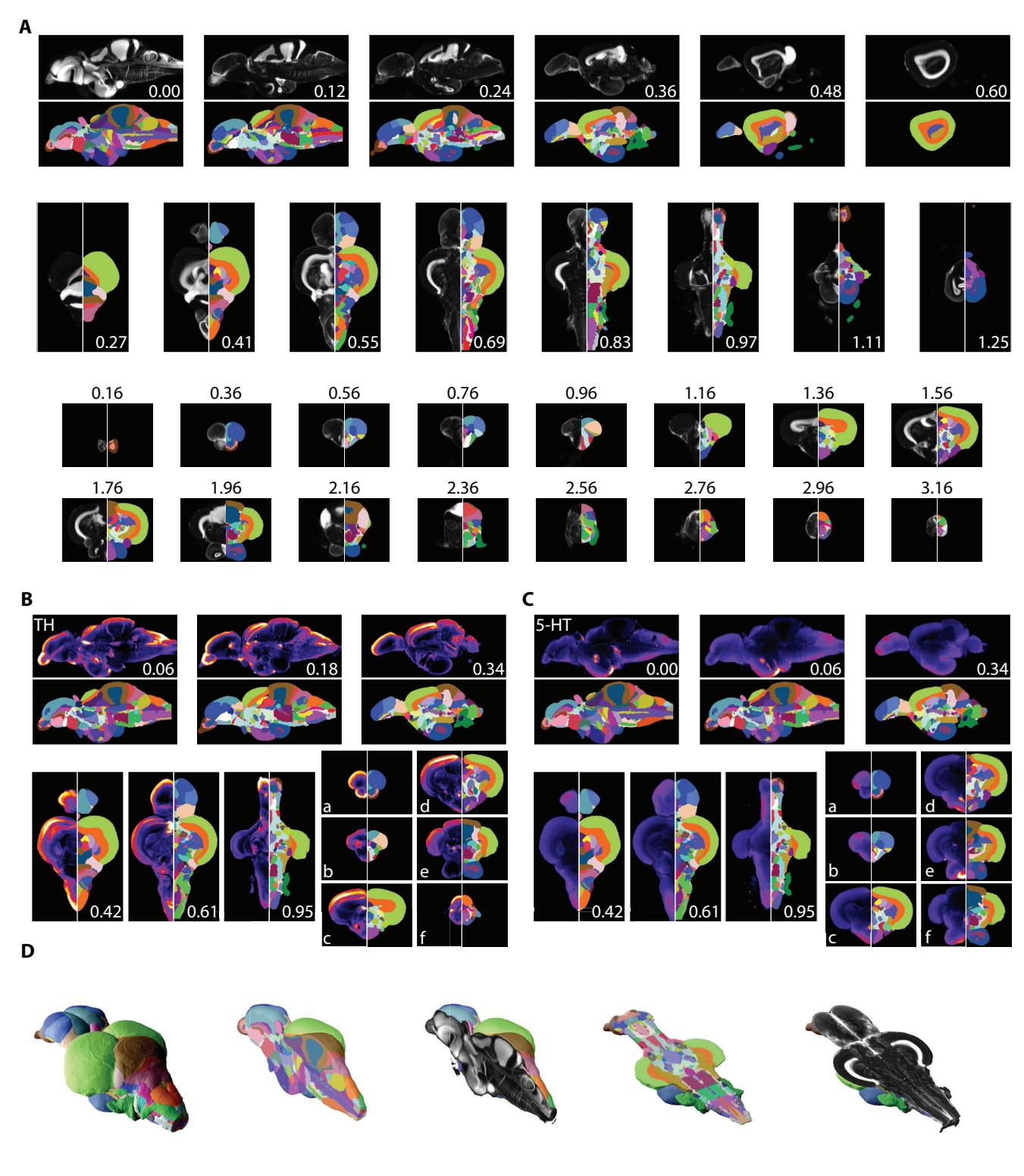

**Figure 5.** Segmentation of the adult zebrafish brain atlas (AZBA). (**A**) Averaged and registered TO-PRO images alongside the atlas segmentation. For sagittal (top), horizontal (middle), and coronal (bottom) planes, numbers are distance (in mm) from the midline, top, and anterior-most portion of the brain, respectively. (**B, C**) Averaged and registered TH and 5-HT-stained images where hotter colors indicate a stronger signal. Numbers same as in (**A**). Slices in each plane were chosen to show regions containing high levels of staining (see Results for description). (**D**) Three-dimensional representation of the segmentation with a sagittal and horizontal cutaway overlaid with the TO-PRO stain of the atlas.

*Figure 5 continued on next page*

*Figure 5 continued*

The online version of this article includes the following figure supplement(s) for figure 5:

**Figure supplement 1.** Averaged and registered antibody stains with corresponding segmentation for antibody stains.

**Figure supplement 2.** Whole-brain volumes of female and male fish used in the TO-PRO registration.

**Figure supplement 3.** Brain structure volumes of female and male fish used in the TO-PRO registration.

work will aid in filling these unsegmented regions with the potential to discover new neuronal circuits and anatomical structures.

Segmentation was also guided by stains with antibodies that have been used in prior studies of the adult zebrafish brain. This includes amines like TH and 5-HT (*Kaslin and Panula, 2001*; *Rink and Wullimann, 2001*), ChAT to identify cholinergic neurons and motor nuclei (*Mueller et al., 2004*), and calcium binding proteins like calretinin (*Castro et al., 2006b*; *Castro et al., 2006a*). All of our immunostainings showed high overlap with prior work, establishing the validity of these antibodies (*Figure 5B and C*, *Figure 5—figure supplement 1*). For example, in the telencephalon we found extensive TH expression in the olfactory bulb (*Figure 5Ba*), and as a continuous band ventral to the Dc region (*Figure 5Bb*). As previously reported, TH staining was also elevated in the optic tectum, thalamic, and hypothalamic regions (*Figure 5Bc,d*), the LC (*Figure 5Be*), and in the XLo of the hind-brain (*Figure 5Bf*). We found 5-HT staining to be more diffuse in regions like the optic tectum and the dorsal part of the telencephalon (*Figure 5Ca,b*), with pockets of high expression restricted to regions like the PVO (*Figure 5Cc*), PTN (*Figure 5Cd*), posterior portion of the Hc (*Figure 5Ce*), and the SR (*Figure 5Cf*). Using ChAT, we could clearly discern staining in places such as the RT (*Figure 5—figure supplement 1Ba*), near the TTB (*Figure 5—figure supplement 1Bb*), and in the SRN and NLV (*Figure 5—figure supplement 1Bc*). ChAT staining was also apparent in several motor nuclei such as the OENr and VIIm (*Figure 5—figure supplement 1Bd*), OENc (*Figure 5—figure supplement 1Be*), and IXm (*Figure 5—figure supplement 1Be*). Finally, examples of high levels of calretinin staining can be seen in the olfactory bulb (*Figure 5—figure supplement 1Ca*), the Dm (*Figure 5—figure supplement 1Cb*), the PSP and optic tract (*Figure 5—figure supplement 1Cc*), optic tectum (*Figure 5—figure supplement 1Cc–e*), the TLa, SG, TGN, and anterior portion of the DIL (*Figure 5—figure supplement 1Cd*) with particularly strong staining in the SGN (*Figure 5—figure supplement 1Ce*) and DON (*Figure 5—figure supplement 1Cf*).

Our segmentation resulted in a 3D model of the zebrafish brain that can be viewed from any arbitrary angle (*Figure 5D*). Each nucleus, white matter tract, ventricle, and anatomical space was given a unique abbreviation and color, totaling 203 regions (*Supplementary file 1*) and associated with an anatomical hierarchy (*Supplementary files 2 and 3*). The full extent of the atlas can be appreciated using the website (azba.wayne.edu) or ITK-SNAP (*Yushkevich et al., 2019*), a freely available software package designed for viewing 3D medical images that allows for the simultaneous viewing of the stains and segmentation in the coronal, sagittal, and horizontal planes (*Figure 6*). All files for exploring AZBA are freely available for use in ITK-SNAP or other programs (https://doi.org/10.5061/dryad.dfn2z351g).

## Sex and brain volume

Because the goal of the atlas was to generate a representative brain, we combined images from both male and female fish. To determine if there was an effect of sex on brain volumes in our TO-PRO image set, we used a mixed-model ANOVA (2 × 203; sex [between subjects] × brain region [within subjects]). We found neither an effect of sex ($F_{(1,15)} = 0.11$, $p=0.75$; *Figure 5—figure supplement 2*) nor an interaction between sex and brain region ($F_{(202, 3030)} = 0.14$, $p=1$; *Figure 5—figure supplement 3*), suggesting that sex did not affect the overall size of the brain or individual regions. We did find a main effect of region ($F_{(202, 3030)} = 495$, $p<2 \times 10^{-16}$), consistent with the wide range of region sizes observed across the brain ($0.000026–0.53$ mm$^3$; *Supplementary file 2*).

## Neurochemical organization of the adult zebrafish brain

We used AZBA to generate insight into the neurochemical organization of the adult zebrafish brain using antibody stains that have not previously been subject to brain-wide examination. Parvalbumin (PV) is a calcium binding protein that labels a class of inhibitory interneurons (*Celio, 1986*). We found

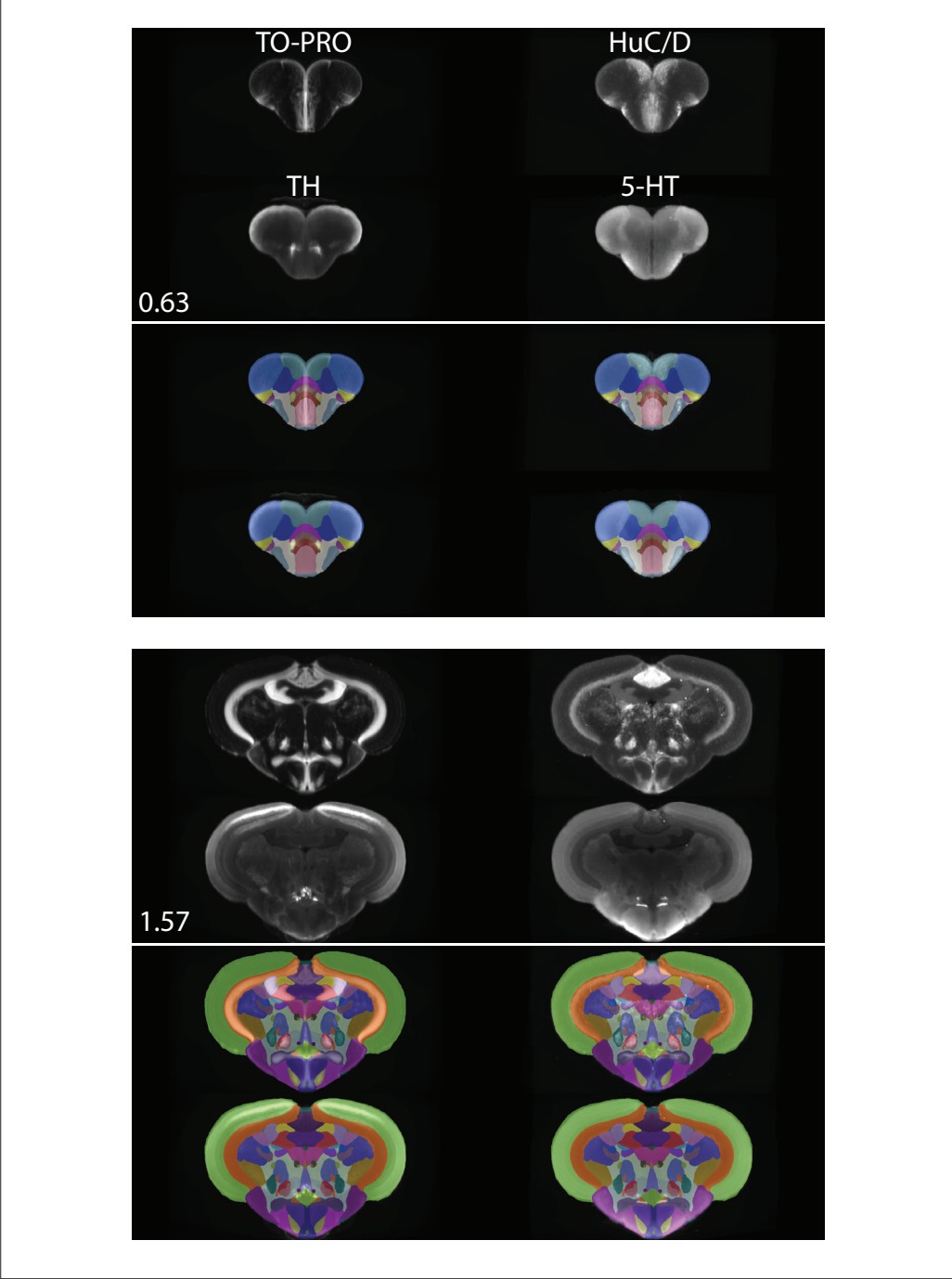

**Figure 6.** Registered and averaged TO-PRO, HuC/D, TH, and 5-HT images in the coronal plane alongside atlas segmentation at 50% opacity and visualized using ITK-SNAP. Numbers are distance (in mm) from the anterior-most portion of the brain.

several highly concentrated areas of PV staining such as in the anterior portions of the olfactory bulb (*Figure 7A*), APN and DAO (*Figure 7Ab*), the TSc (*Figure 7Ac*), the DON and VIII of the hindbrain (*Figure 7Af*), and very high levels in the ventral portion of the molecular layer of the cerebellum (*Figure 7Ad,e*) the latter of which likely corresponding with Purkinje and crest cells (*Bae et al., 2009*).

Prox1 is a homeobox gene critical for regulating neuronal development with widespread expression in larval fish. In juveniles and adult zebrafish, Prox1 expression decreases rapidly and is eventually confined to a relatively few regions (*Ganz et al., 2012*; *Pistocchi et al., 2008*). In the adult zebrafish

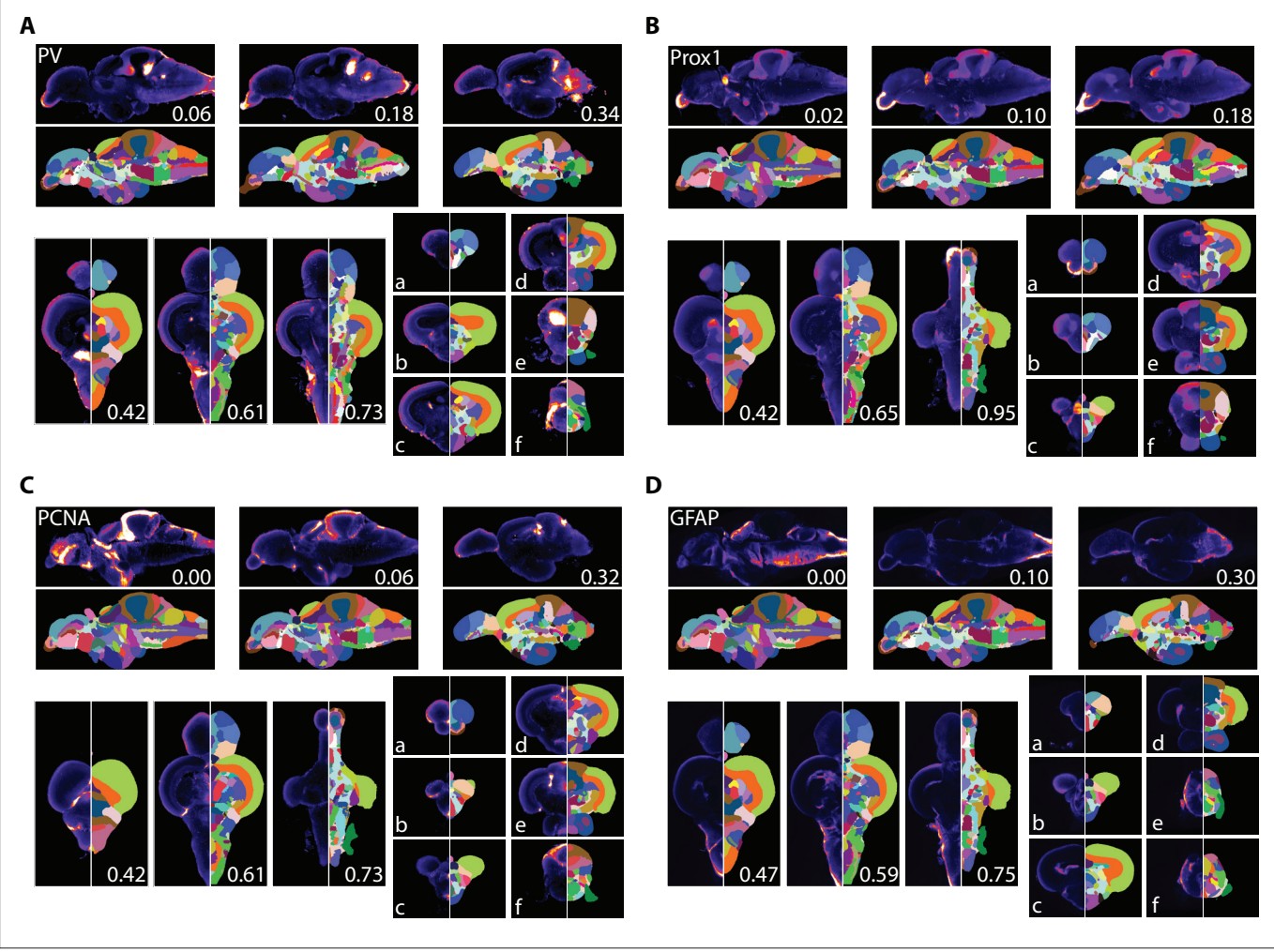

**Figure 7.** Averaged and registered antibody stains with corresponding segmentation for (**A**) PV, (**B**) Prox1, (**C**) PCNA, and (**D**) GFAP where hotter colors indicate greater staining. For sagittal (top) and horizontal (left, bottom), numbers represent distance (in mm) from midline or top of the brain, respectively. Slices for each plane were chosen based on the presence of staining (see Results for description).

pallium, Prox1 was reported to be present in the neuronal layer of the Dl with more diffuse staining in posterior portion of the Dc, which we also observed (*Figure 7Bb*). We also observe high levels of Prox1 in the olfactory bulb (*Figure 7Ba*), the habenula (*Figure 7Bb*), the Val and hypothalamus (*Figure 7Bd*), and the DIL and the molecular layer of the cerebellum (*Figure 7Be,f*).

PCNA is a marker for proliferating cells (*Grandel et al., 2006*; *Wullimann and Puelles, 1999*). Consistent with widespread neurogenesis in the adult zebrafish brain, we found PCNA expressed in many neurogenic niches with the highest expression along the midline (*Figure 7C*). For example, in the telencephalon we noticed a band of high expression along the midline in the ventral telencephalon that begins near the olfactory bulbs (*Figure 7Ca*) with a second area in the posterior portion corresponding to the PPa (*Figure 7Cb*). We also saw high levels of expression in parts of the thalamus (VM; *Figure 7Cc*), the hypothalamus (Hd and posterior portion of Hc) and the valvula, caudal lobe, and molecular layers of the cerebellum (*Figure 7Cd–f*).

Calbindin is a calcium binding protein important for regulating intracellular signaling that is often used in comparative neurological studies (*Schmidt, 2012*). We found calbindin to be concentrated in the fiber layers of the olfactory bulbs (*Figure 5—figure supplement 1Da*), the Dm in the telencephalon (*Figure 5—figure supplement 1Db*), the VOT (*Figure 5—figure supplement 1Dc*), throughout the optic tectum (*Figure 5—figure supplement 1Dc–e*), the TLa (*Figure 5—figure supplement 1Dd*), SGN (*Figure 5—figure supplement 1De*), and in the VIII and DON of the hindbrain (*Figure 5—figure*

supplement 1Df). Notably, calbindin staining patterns largely overlapped with calretinin (*Figure 5—figure supplement 1C*).

GFAP is a marker for non-neuronal cells like astrocytes, radial glial cells, and ependymal cells (*Doetsch et al., 1997*; *Eng et al., 2000*). Accordingly, we found GFAP most concentrated near the midline and ventricles (*Figure 7D*). Pockets of expression were present that were not adjacent to ventricles such as near the entopeduncular nuclei (*Figure 7Da*), the habenula and anterior portions of the thalamus (*Figure 7Db*), and nuzzled between the NI and TSvl (*Figure 7Dd*). Throughout the hindbrain, GFAP expression was largely restricted to the edges of the brain with the exception of the IR and dorsolateral edge of the XLo (*Figure 7De,f*).

## Discussion

In this article, we introduce a new resource for the zebrafish community: AZBA, a 3D brain atlas for adult zebrafish that can be downloaded (https://doi.org/10.5061/dryad.dfn2z351g) or explored on the web (azba.wayne.edu). This resource will facilitate a wide variety of neurobiological studies using adult zebrafish aimed at dissecting neural circuits of behavior, understanding brain pathology, and discovering novel and conserved neuroanatomy. We created AZBA by leveraging advances in tissue clearing, light-sheet fluorescent microscopy, and image registration, resulting in the most detailed atlas for adult zebrafish to date. Tissue clearing allowed us to take a whole-mount approach, overcoming the natural opacity of the adult brain and issues associated with slice-based techniques such as tissue loss, tearing, and distortion. Laser fluorescence light-sheet microscopy was used to image the large volume of the zebrafish brain with high resolution and minimal photobleaching. Finally, we used 3D image registration to create images derived from multiple animals and inclusion of 10 antibody stains into the same anatomical space. These were then used to guide segmentation of the atlas into over 200 different neuroanatomical regions.

AZBA represents a significant departure from two prior brain atlases for adult zebrafish. The seminal book atlas from *Wullimann et al., 1996* is exceptionally detailed and has guided zebrafish neuroscience research for over two decades. However, being in print, it has not been updated with the latest findings and its two-dimensional visual presentation and lack of chemoarchitectural markers makes identifying regional boundaries across anatomical planes problematic. More recently, Ullmann and colleagues used magnetic resonance imaging (MRI) to create a 3D atlas for adult zebrafish (*Ullmann et al., 2010*). Using a 16.4 Tesla magnet, they imaged a single brain at approximately 10 µm resolution and segmented it into 53 regions. Although an important achievement, MRI is limited in its ability to detect neurochemical markers and does not integrate easily with genetic labeling techniques. Furthermore, such strong magnets are not readily available to most researchers. In contrast, by combining recent advances in widely available tissue clearing techniques, light-sheet microscopy, and image registration, AZBA is highly versatile, detailed, and accessible.

Antibody stains were central to developing AZBA because they validated our approach and improved segmentation through comparison to prior work. Indeed, patterns of our neurotransmitter related stains largely agree with previous reports probing TH (*Castro et al., 2006a*; *Kaslin and Panula, 2001*; *Ma, 2003*; *Yamamoto et al., 2010*), 5-HT (*Kaslin and Panula, 2001*; *Norton et al., 2008*), and ChAT (*Castro et al., 2006a*; *Clemente et al., 2004*; *Mueller et al., 2004*). PCNA also overlaps with earlier findings using both PCNA antibodies and bromodeoxyuridine labeling (*Ampatzis et al., 2012*; *Byrd and Brunjes, 2001*; *Grandel et al., 2006*; *Ito et al., 2010*; *Makantasi and Dermon, 2014*; *von Krogh et al., 2010*). Likewise, stains against the calcium binding proteins like calretinin, calbindin, and PV largely overlap with prior work with only minor exceptions. Consistent with previous studies, we find calretinin expressed in the olfactory bulb (*Kress et al., 2015*), telencephalon (*Porter and Mueller, 2020*), and posterior parts of the brain, with the only notable exceptions being our lack of staining in the torus semicircularis and perilemniscal nucleus (*Castro et al., 2006a*; *Castro et al., 2006b*). For calbindin, despite using the same antibody, we find that our staining in the telencephalon looks different than previous work (*von Trotha et al., 2014*) where we find little staining in the subpallium, and instead see staining in the medial zone of the dorsal telencephalon limited to its posterior extent. For PV, we also note significant overlap with prior work where we see labeling in cell bodies of the cerebellum and olfactory bulbs, but less consistency in the more diffuse staining in the telencephalon (*Ampatzis and Dermon, 2007*; *Bae et al., 2009*; *Mueller et al., 2011*; *Porter and Mueller, 2020*). Some of these minor discrepancies may be due to changes in antigenicity arising

from the use of different methodologies for fixation, loss of sparse signal due to the averaging of multiple images, tissue distortion (*Renier et al., 2016*; *Richardson and Lichtman, 2015*), or because prior work reported images from single animals and may be more susceptible to individual variation in expression patterns compared to the present work where images are averaged across several subjects. Nonetheless, our high correspondence with the previous literature suggests that the present work accurately represents the adult zebrafish brain.

Of the antibody stains that had not been subject to extensive prior work, the findings of greatest interest are from Prox1, a homeobox protein critical to the development of an array of organs and cell types including neurons during embryonic and adult stages (*Elsir et al., 2012*; *Kaltezioti et al., 2010*; *Karalay et al., 2011*). We found Prox1 staining in the telencephalon, consistent with prior work (*Ganz et al., 2012*), as well as the habenula, parts of the hypothalamus, and the cerebellum (*Figure 7B*). Hypothalamic *Prox1* expression in larval zebrafish has been found to be important for the development of catecholaminergic neurons (*Pistocchi et al., 2008*). However, we do not see overlap between TH and Prox1 in the hypothalamus, suggesting that Prox1 may be important for the development, but not the maintenance, of hypothalamic catecholaminergic neurons. In adult mice, Prox1 is present in the dentate gyrus of the hippocampus and GFAP-positive cells localized to white matter tracts in the cerebellum (*Karalay et al., 2011*; *Lavado and Oliver, 2007*). Interestingly, we find overlap between Prox1 and GFAP in the olfactory bulb and the edges of the telencephalon, but not the cerebellum, where Prox1 has greater overlap with PCNA, a marker for proliferating cells. The dentate gyrus in mice is notable for being one of the few areas where adult neurogenesis has been demonstrated (*Ming and Song, 2011*). Thus, the broader expression of Prox1 in our study may reflect the presence of more widespread neurogenesis in adult zebrafish compared to mice (*Kizil et al., 2012*).

We view the current segmentation and image collection that comprises AZBA as a first version that will be continually updated. To facilitate updating, and encourage input from the scientific community, we have created a website (azba.wayne.edu) where we invite comments and suggestions for updates. In addition, we expect future work will incorporate more antibody images and in situ hybridization probes for understanding how patterns of protein and gene expression vary across the brain. Through collaboration with the zebrafish community, we plan to incorporate the wealth of Gal4 and Cre/loxP lines that have been generated to characterize expression patterns in the adult brain. We envision a process where scientists send fixed brain samples to a central lab for tissue clearing, imaging, and registration to the atlas for incorporation into our online resource. A similar approach has been taken with larval fish (*Kawakami et al., 2010*; *Kunst et al., 2019*; *Randlett et al., 2015*; *Ronneberger et al., 2012*; *Tabor et al., 2019*). However, images from adult animals are important because transgene expression patterns can change as animals mature (*Lal et al., 2018*).

AZBA also enables new insight into the functional organization of the zebrafish brain by facilitating whole-brain activity mapping as has been achieved with larval zebrafish (*Ahrens et al., 2012*; *Randlett et al., 2015*). New images can be automatically segmented into individual brain regions by registration to our averaged autofluorescence or nuclear-stained images (*Figure 4*). Because adult zebrafish have a mature neuroanatomy and a larger behavioral repertoire than larval fish, this will provide an important new avenue for exploiting the power of the zebrafish model system to yield insight into the functional organization of the vertebrate brain and how it relates to behavior.

AZBA provides an unprecedented view of the adult zebrafish brain, consisting of averaged 3D nuclear-stained and antibody images registered into the same space. All files associated with the atlas are available to the community (https://doi.org/10.5061/dryad.dfn2z351g) and can be viewed online (azba.wayne.edu) or with ITK-SNAP, a freely available software package (*Yushkevich et al., 2019*). With AZBA, adult zebrafish join the ranks of other vertebrate model organisms in neuroscience that have highly detailed digital atlases such as larval zebrafish (*Kunst et al., 2019*; *Randlett et al., 2015*; *Ronneberger et al., 2012*; *Tabor et al., 2019*), mice (*Lein et al., 2007*), and rhesus macaques (*Reveley et al., 2017*). We anticipate this resource will contribute to the ascent of adult zebrafish into the upper echelons of model organisms in neuroscience and facilitate our understanding of the evolution, development, and functioning of the vertebrate central nervous system.

## Materials and methods

**Key resources table**

| Reagent type (species) or resource | Designation | Source or reference | Identifiers | Additional information |
|---|---|---|---|---|
| Strain, strain background (*Danio rerio*) | Zebrafish (AB strain) | The Hospital for Sick Children | | |
| Other | TO-PRO3 stain | Invitrogen | Cat#: T3605 | iDISCO (1:10,000) |
| Antibody | Anti-TH (chicken polyclonal) | Aves | Cat#: TYH; RRID:AB_10013440 | iDISCO (1:200) |
| Antibody | Anti-GFAP (mouse monoclonal) | ZIRC (Zebrafish International Resource Center) | Cat#: Zrf-1 | iDISCO (1:200) |
| Antibody | Anti-ChAT (goat polyclonal) | Millipore | Cat#: AB144; RRID:AB_11212843 | iDISCO (1:400) |
| Antibody | Anti-5-HT (rabbit polyclonal) | Sigma | Cat#: S5545; RRID:AB_477522 | iDISCO (1:100) |
| Antibody | Anti-calbindin (rabbit polyclonal) | SWANT | Cat#: CB38; RRID:AB_10000340 | iDISCO (1:200) |
| Antibody | Anti-calretinin (mouse monoclonal) | SWANT | Cat#: 6B3; RRID:AB_10000320 | iDISCO (1:400) |
| Antibody | Anti-PCNA (mouse monoclonal) | Dako | Cat#: M0879; RRID:AB_2160651 | iDISCO (1:1000) |
| Antibody | Anti-HuC/D (mouse monoclonal) | Invitrogen | Cat#: A21271; RRID:AB_221448 | iDISCO (3.75 µg/mL) |
| Antibody | Anti-Prox1 (rabbit polyclonal) | Millipore | Cat#: AB5475; RRID:AB_177485 | iDISCO (1:400) |
| Antibody | Anti-parvalbumin (rabbit polyclonal) | SWANT | Cat#: PV27; RRID:AB_2631173 | iDISCO (1: 400) |
| Antibody | Anti-mouse IgG-Alexa Flour 647 (donkey polyclonal) | Invitrogen | A31571 RRID:AB_162542 | iDISCO (1:200) |
| Antibody | Anti-rabbit IgG-Alexa Flour 647 (donkey polyclonal) | Invitrogen | A31573 RRID:AB_2536183 | iDISCO (1:200) |
| Antibody | Anti-goat IgG-Alexa Flour 647 (donkey polyclonal) | Invitrogen | A21447 RRID:AB_2535864 | iDISCO (1:200) |
| Antibody | Anti-chicken-Alexa Flour 633 (goat polyclonal) | Invitrogen | A21103 RRID:AB_2535756 | iDISCO (1:200) |
| Other | AZBA | This paper | RRID:SCR_021732 | Brain atlas maintained by J.W. Kenney lab Available at azba.wayne.edu |

## Subjects

Subjects were AB fish (15–16 weeks of age) of both sexes. Fish were housed in 2 L tanks with 8–12 fish per tank. All fish were bred and raised at the Hospital for Sick Children in high-density racks with a 14:10 light/dark cycle (lights on at 8:30) and fed twice daily with *Artemia salina*. All procedures were approved by the Hospital for Sick Children Animal Care and Use Committee.

## Sample preparation

Zebrafish were euthanized by anesthetizing in 4% tricaine followed by immersion in ice-cold water for 5 min. Animals were then decapitated using a razor blade and heads were placed in ice-cold PBS for 5 min to let blood drain. Heads were then fixed in 4% PFA overnight after which brains were then carefully dissected into cold PBS and stored at 4°C until processing for iDISCO+. Brains that were damaged during the dissection process were not used for generating the atlas.

## Tissue staining

Tissue staining and clearing was performed using iDISCO+ (*Renier et al., 2016*). Samples were first washed three times in PBS at room temperature, followed by dehydration in a series of methanol/ water mixtures (an hour each in 20, 40, 60, 80, and 100% methanol). Samples were further washed in

100% methanol, chilled on ice, and then incubated in 5% hydrogen peroxide in methanol overnight at 4°C. The next day samples were rehydrated in a methanol/water series at room temperature (80, 60, 40, and 20% methanol) followed by a PBS wash and two 1 hr washes in PTx.2 (PBS with 0.2% TritonX-100). Samples were then washed overnight at 37°C in permeabilization solution (PBS with 0.2% TritonX-100, 0.3 M glycine, 20% DMSO) followed by an overnight incubation at 37°C in blocking solution (PBS with 0.2% TritionX-100, 6% normal donkey serum, and 10% DMSO). Samples were then labeled with TO-PRO3 iodide (TO-PRO) (one night) or primary antibodies (2–3 nights) via incubation at 37°C in PTwH (PTx.2 with 10 µg/mL heparin) with 3% donkey serum and 5% DMSO. Samples were then washed at 37°C for 1 day with five changes of PTwH. Antibody-stained samples were followed by incubation with secondary antibodies at 37°C for 2–3 days in PTwH with 3% donkey serum. For samples labeled with TO-PRO, the secondary antibody labeling step was omitted. Following secondary antibody labeling, samples were again washed at 37°C in PTwH for 1 day with five solution changes.

## Tissue clearing

Labeled brains were first dehydrated in a series of methanol water mixtures at room temperature (an hour each in 20, 40, 60, 80, and 100% [×2] methanol) and then left overnight in 100% methanol. Samples were then incubated at room temperature in 66% dichloromethane in methanol for 3 hr followed by two 15 min washes in dichloromethane. After removal of dichloromethane, samples were incubated and stored in dibenzyl ether until imaging.

## Imaging

All imaging was done on a LaVision ultramicroscope I. Samples were mounted using an ultraviolet curing resin (adhesive 61 from Norland Optical, Cranbury, NJ) that had a refractive index (1.56) that matched the imaging solution, dibenzyl ether. Images were acquired in the horizontal plane at 4× magnification.

## Image processing

Datasets from light-sheet imaging were stitched using Fiji's (NIH) extension for Grid Stitching (*Preibisch et al., 2009*) and converted to a single stack, corresponding to the z-axis. All image processing steps were run on a Linux workstation with 64 GB of RAM and 12-core Intel processor.

Each stack was converted to a 4 µm isotropic image as previously described (*Vousden et al., 2014*; *Supplementary file 4*) with separate files for the autofluorescence channel and a second for the antibody or TO-PRO channels. These images were resampled to 8 µm isotropic due to system constraints during the image registration stages.

## Registration

The TO-PRO and autofluorescence signals were acquired on an initial dataset of 17 fish. To create the initial average, we used image registration to align in a parallel groupwise fashion the TO-PRO images. The variability was expected to be less in the TO-PRO because these images contained more contrast than the autofluorescence images.

The creation of an initial average of the adult zebrafish brain was accomplished using 17 samples with the TO-PRO channel. The process was completed using a three-step registration process, similar to prior work (*Lerch et al., 2011*) using the pydpiper pipeline framework (*Friedel et al., 2014*) and the minctracc registration tool (*Collins and Evans, 2011*). This involved taking a single sample at random and registering the 17 samples to it using a six-parameter linear alignment process (LSQ6). This yielded 17 samples in similar orientation to allow a 12-parameter linear registration (LSQ12) to be performed in a pairwise fashion (each sample is paired with all the other samples to avoid sample bias), and the final output of these 12-parameter registration was a group average. This represents a linearly registered average adult zebrafish brain. This was then used as the target for nonlinear registration with each of the linearly registered 17 TO-PRO samples. This nonlinear alignment was repeated successively with smaller step sizes and blurring kernels to allow for an average with minimal bias from any one sample brain. We then took this average and mirrored itself along the long axis of the brain and repeated the registration process described above, but instead of using a random brain as the six-parameter target, we used this mirrored brain. The result of this second pipeline was an average brain where each plane of the brain (coronal, sagittal, horizontal) is parallel with the

imaging planes (x, y, z). This final average brain represented the starting point of the atlas. The linear and nonlinear transformations created in the registration pipeline were used to resample the 4 µm isotropic TO-PRO and autofluorescence images to the atlas space, yielding an average signal for each channel. The autofluorescence signal was used to register other sample datasets with the atlas because it is common across all datasets.

To combine the additional cellular markers to better delineate structures and examine their distribution across the brain, we converted all images and their channels to 4 µm isotropic images as described above. We then converted them to 8 µm isotropic and used the autofluorescence channel for each set to run the above registration pipeline (LSQ6, LSQ12, and nonlinear). The initial target was the autofluorescence average created with the TO-PRO dataset described above. Following each registration pipeline, the transformations were used to resample each autofluorescence and cellular marker channel to the atlas with a resolution of 4 µm isotropic.

To assess registration precision using TO-PRO or autofluorescence images, for each signal we identified six landmarks in the atlas and their corresponding location on 7-8 different image sets. These points were then brought into atlas space using the transformations from the registration process. We then computed the Euclidean distance between the points in the atlas image and the transformed images for the TO-PRO and autofluorescence signals. Precision data are presented as mean ± standard deviation unless otherwise indicated.

## Segmentation

Segmentation was performed using ITK-SNAP, a freely available software package for working with multimodal medical images that enables side-by-side viewing of 3D images registered into the same anatomical space (*Yushkevich et al., 2019*). Segmentation was primarily guided by comparing TO-PRO nuclear-stained images to the cresyl violet stain of the original atlas (*Wullimann et al., 1996*). Boundaries of nuclei were often determined using the TO-PRO stain in conjunction with a neuronal marker (HuC/D) and other antibody stains as needed. Terminology largely follows that of the original atlas with the exception of motor nuclei (*Mueller et al., 2004*) and the telencephalon (*Porter and Mueller, 2020*).

## Statistical analysis

Statistical analysis was performed in R (version 4.0.2) using an independent-samples t-test or mixed-model ANOVA, as indicated.

## Acknowledgements

We thank Maria Palazzolo and Kailyn Fields for help with preparing files associated with the atlas. We thank Angela Morley, Alan Ng, Monica Yu, and Hillary Winstanley for excellent care of the zebrafish, and thank Tod Thiele for helpful comments on the manuscript. This work was supported by the Human Frontiers Science Program (HFSP; LT000759/2014), Wayne State University Start-up funds, and the National Institutes of Health (NIH; R35GM142566) to JWK, the Canadian Institute for Health Research to PWF (FDN143227). PWF is a senior fellow in the Canadian Institute for Advanced Research program in Child and Brain Development. TM was supported by the Cognitive and Neurobiological Approaches to Plasticity Center, a Center of Biomedical Research Excellence of the NIH (P20GM113109), and by the HFSP (RGP0016/2019).

## Additional information

### Funding

| Funder | Grant reference number | Author |
| --- | --- | --- |
| Human Frontier Science Program | LT000759/2014 | Justin W Kenney |
| National Institutes of Health | R35GM142566 | Justin W Kenney |

| Funder | Grant reference number | Author |
|---|---|---|
| Canadian Institute for Health Research | FDN143227 | Paul W Frankland |
| National Institutes of Health | P20GM113109 | Thomas Mueller |
| Human Frontier Science Program | RGP0016/2019 | Thomas Mueller |

The funders had no role in study design, data collection and interpretation, or the decision to submit the work for publication.

## Author contributions

Justin W Kenney, Conceptualization, Data curation, Formal analysis, Funding acquisition, Investigation, Methodology, Project administration, Resources, Supervision, Validation, Visualization, Writing – original draft, Writing – review and editing; Patrick E Steadman, Conceptualization, Data curation, Investigation, Methodology, Software, Visualization, Writing – original draft, Writing – review and editing; Olivia Young, Meng Ting Shi, Maris Polanco, Saba Dubaishi, Investigation; Kristopher Covert, Resources, Software, Visualization; Thomas Mueller, Investigation, Validation, Writing – review and editing; Paul W Frankland, Funding acquisition, Supervision, Writing – review and editing

## Author ORCIDs

Justin W Kenney http://orcid.org/0000-0001-8790-5184
Paul W Frankland http://orcid.org/0000-0002-1395-3586

## Ethics

The study was performed in accordance with the Guide for the Care and Use of Laboratory Animals of the National Institutes of Health. All procedures were approved by the animal care committee of The Hospital for Sick Children (protocol #0000047792).

## Decision letter and Author response

Decision letter https://doi.org/10.7554/eLife.69988.sa1
Author response https://doi.org/10.7554/eLife.69988.sa2

# Additional files

## Supplementary files

• Supplementary file 1. Table of brain region abbreviations, full names, and colors.

• Supplementary file 2. Excel file of brain region label numbers, abbreviations, full names, colors, volume, hierarchy, and location.

• Supplementary file 3. Structure tree of brain region hierarchies.

• Supplementary file 4. Python script for converting image stacks to three-dimensional volumes.

• Transparent reporting form

## Data availability

Data have been deposited in Dryad, accessible at: https://doi.org/10.5061/dryad.dfn2z351g.

The following dataset was generated:

| Author(s) | Year | Dataset title | Dataset URL | Database and Identifier |
|---|---|---|---|---|
| Kenney JW, Steadman P, Young O, Shi M, Polanco M, Dubaishi S, Covert K, Mueller T, Frankland P | 2021 | Data from: A 3D Adult Zebrafish Brain Atlas (AZBA) for the Digital Age | https://doi.org/10.5061/dryad.dfn2z351g | Dryad Digital Repository, 10.5061/dryad.dfn2z351g |

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
