## [Editor Report]

The issue of building a high-quality brain atlas for vertebrates has been a long-standing challenge in the field. Your work has nicely hit this mark using zebrafish, and using a method that should be applicable to many different stages and other organisms.

---

## [Decision Letter]

**Decision letter after peer review:**

Thank you for submitting your article "AZBA: A 3D Adult Zebrafish Brain Atlas for the Digital Age" for consideration by *eLife*. Your article has been reviewed by 3 peer reviewers, including Stephen C Ekker as Reviewing Editor and Reviewer #1, and the evaluation has been overseen by Marianne Bronner as the Senior Editor. The following individual involved in review of your submission has agreed to reveal their identity: Harold Burgess (Reviewer #2).

Essential revisions:

The following issues need to be addressed, and should not cause much of a delay:

1) Non-annotated areas should not be "0" like areas outside (use of black in both cases).

2) There should be one table containing colors, abbreviations and full names of anatomical structures (which will make the data much more and easier accessible). Assigning all anatomical entities to larger brain structures (telencephalon etc) may not be possible for all structures due to anatomical disputes, but also very helpful where possible.

3) Think they should do an analysis to at least show the differences are not so huge that mixing the sexes is reasonable.

*Reviewer #1 (Recommendations for the authors):*

This paper uses cutting-edge imaging to develop a new 3D map of the zebrafish brain. The use of fixed imaging plus antibody staining with Lightsheet microscopy has developed an excellent high-resolution dataset. The regional imaging data is convincing. The data is well-presented, and the text easy to read. This is an ideal manuscript within the broad scope of *eLife*.

A few questions remain to better understand the outcomes they describe:

1) For smaller brain regions, how do they validate their annotation? This was not clear.

2) What is the true estimate of error of their imaging approach? 8 microns is pretty large, and might lead to errors in tagging individual cells. How do they sort this out? This is not well presented in this current version of the manuscript.

3) How many different fish did they image? We now know there are a LOT of differences even between siblings of wild-type lines. This would provide another form of error they would need to address, if nothing else for users to explore their resource.

*Reviewer #2 (Recommendations for the authors):*

My comments below should be interpreted only as efforts to make this atlas as user-friendly as possible.

Understandably, due to the incomplete knowledge of the zebrafish brain, some parts of the brain have not been segmented. At present, these areas are indexed as 0 (zero), which is the same as the area outside the brain. I would strongly encourage the authors to distinguish unannotated voxels from space outside of the brain, as this will be essential to facilitate computational analyses of brain imaging data.

I suspect that the annotation file "2021-05-02_AZBA_Label_descriptions.txt" is formatted for ITK-SNAP. However, it could be more useful for other computational studies:

(1) At present, the file contains only region abbreviations, making it necessary to constantly consult Table S2 of the manuscript. I would like this file to additionally include the full name of the region.

(2) Each region should be associated with a major brain division (telencephalon, diencephalon etc) and where possible, given any other useful annotations (ventricle, tract, nucleus). These labels provide valuable grist for computational methods.

(3) For people not using ITK-SNAP it would be helpful to provide the coordinates of a voxel within each region. For example I tried in vain to hunt for the voxels corresponding to MAC.

(4) I understand the first column (index) and last column (abbreviation). What are the other columns?

After light-sheet imaging, what actually happens during the step referred to as "3D volume generation", or "Three-dimensional volumes were created from individual image sets", or "Image stacks from individual fish brains were converted to 3D volumes”? Does this simply refer to the stitching plus resampling procedure?

If I understand correctly, the ultimate step in generating a template brain was to mirror the first average, then register the previously linearly aligned TOPRO samples to to it. Although I appreciate from the data I retrieved from Dryad that the result is a symmetric brain, I can't quite understand why this procedure yielded a symmetric template, rather than just a more precise template with left/right reversal, so please describe what happened in this step more clearly.

More importantly, while in general a symmetric template is desirable (especially for computational approaches), this procedure obscures important biological asymmetries present in the habenula, a major focus of neurobiological studies in zebrafish. Could you somehow exclude the habenula from the symmetrization step?

No information is provided on the precision of the registration procedure. It would be helpful to assess precision so that subsequent users can evaluate whether their own procedures provide sufficiently close alignments.

I expect that autofluorescence is more variable than TOPRO and carries less information. However there is no information provided for evaluating the quality of the registration achieved using autofluorescence as a bridging channel. Figure 4D (which is small and hard to see) purports to illustrate the accuracy of autofluorescence bridging registration using the example of the Locus Coeruleus. However, because one can not reliably ascertain the LC in either the HuC or TOPRO stain, there is nothing to usefully compare the TH stain to. Precise registration is important because the IHC channels used for segmentation were registered using autofluorescence. Please provide a measurement of registration accuracy using autofluorescence. Why was TOPRO not used with IHC for this step, to provide the strongest alignment?

The Methods are generally appropriately detailed, and specify the age of the fish. However, 3-4 month old fish can vary widely in size, and therefore presumably in brain volume. It may be helpful to add details on the size of the fish, and how sensitive the accuracy of the registration procedure is to size.

Much of the segmentation was performed on coronal sections, for which region boundaries are generally smooth. However, when viewed in other planes, boundaries are jagged resulting in artificial discontinuities within regions. It would be desirable to implement at least a simple procedure to smooth region boundaries in 3D space.

*Reviewer #3 (Recommendations for the authors):*

In the intro page 4 the authors claim the adult zebrafish brain would be "several orders of magnitude larger" than the larval brain. "Several" to me means more than two, but I would think that the adult zebrafish brain is just one order of magnitude larger (larva 500 micrometer, adult 5-15 mm).

Some images, e.g. in Figure 4, especially 4d, but also parts of Figure 7, are provided with too low resolution in the PDF to judge detail.

Figure 7B middle panels "0,10"

Prox1 appears to be an example that a major expression domain maps to a non-annotated anatomical domain – or the prethalamus is so dark here that it cannot be distinguished from black. In the text where Prox1 is presented, this domain appears to be not mentioned.

For some panels it may be better to present cutouts at higher magnification, with anatomical regions outlined in fine white lines and labelled (rather than using the color code only).

D GFAP: I cannot identify in the figure the GFAP pattern described in the text:

"we found GFAP most concentrated near the midline and ventricles (Figure 7D)."

Why is there no GFAP signal in the midsagital section 7D 0.00?

Materials:

Given sex differences in the fish brain (see papers cited by authors), the authors should report whether male or female brains or a mix of brains from both sexes were used.

Limitations

Given that relatively few antibodies are available for specific zebrafish neuronal markers, it would be very informative to know if the technique used by the authors would in principle be applicable to fluorescent whole mount hybridization stained brains.

---

## [Author Response]

Essential revisions:The following issues need to be addressed, and should not cause much of a delay:1) Non-annotated areas should not be "0" like areas outside (use of black in both cases).

Non-annotated areas within the atlas are now labelled as “Unknown” with reference to the most relevant ontological level (e.g. “Unknown Ventral Telencephalon (UnkVT), Unknown Diencephalon (UnkD) etc.). These unknown regions have also been given label numbers starting with 900. They are now included in the updated segmentation as can be seen in the figures, the Supplementary files, the website, and the files available on Dryad. We also mention them on lines 227-234.

2) There should be one table containing colors, abbreviations and full names of anatomical structures (which will make the data much more and easier accessible). Assigning all anatomical entities to larger brain structures (telencephalon etc) may not be possible for all structures due to anatomical disputes, but also very helpful where possible.

We have now included a supplemental Excel file to capture all the additional information (Supplementary file 2). We have also included coordinates to help users find a given brain region (in response to a comment from reviewer #2 below) and the volumes of each brain region. To help visualize and navigate the ontological hierarchy, we have also included a structure tree (Supplementary file 3).

3) Think they should do an analysis to at least show the differences are not so huge that mixing the sexes is reasonable.

We have now performed an analysis comparing the brain volumes of male and female fish that were used in the TO-PRO registration. We found no effect of sex on the overall size of the brains and no interaction between sex and brain region. We have included a discussion of this data on lines 269-278 and included two additional figures (Figure 5 —figure supplements 2 and 3).

Reviewer #1 (Recommendations for the authors):This paper uses cutting-edge imaging to develop a new 3D map of the zebrafish brain. The use of fixed imaging plus antibody staining with Lightsheet microscopy has developed an excellent high-resolution dataset. The regional imaging data is convincing. The data is well-presented, and the text easy to read. This is an ideal manuscript within the broad scope of eLife.A few questions remain to better understand the outcomes they describe:1) For smaller brain regions, how do they validate their annotation? This was not clear.

We used several approaches to determine the boundaries of smaller regions. Our primary source of information was the original Wullimann (1996) atlas; notably, our nuclear stain (TO-PRO) yields similar images to the Nissil stain of the Wullimann atlas (lines 143-144). We combined a close examination of this atlas in all axes with recent literature and the fact that we could visualize multiple stains in the same anatomical space at once using ITK-SNAP. By way of example, we show the locus coeruleus (Figure 4D), where we used the TO-PRO, TH, and HuC/D stains to determine the boundary. Another example would be the superior raphe where we used TO-PRO, HuC/D, and 5HT stains to identify the boundaries. The boundaries for many other regions lacking more specific stains were done by combining the TO-PRO and HuC/D stains; we have added sentences to clarify this last point (lines 223-225). However, there was no one combination of stains and approaches that would work for every small region, so we would explore various stains/views/literature to identify boundaries while trying to maintain a smooth segmentation. We also worked with Thomas Mueller, a world renowned expert in zebrafish neuroanatomy, and an author on the manuscript, on segmentation. Nonetheless, we recognize that there may be disagreement with some of our choices for boundaries, and thus we anticipate updating and continually improving the atlas as we incorporate more stains and obtain feedback from the scientific community. To facilitate this, we have now included a website (azba.wayne.edu) that includes contact information for the lead π for any comments users may have on the atlas (lines 397-399).

2) What is the true estimate of error of their imaging approach? 8 microns is pretty large, and might lead to errors in tagging individual cells. How do they sort this out? This is not well presented in this current version of the manuscript.

When creating the atlas our goal was not to identify individual cells but to be able to delineate the boundaries of different brain regions (we have clarified this point by altering lines 146-147 to make clear that our goal was to identify regional boundaries not individual cells). For registration, we downsampled images to 8 micron isotropic resolution because higher resolution required more computational resources than we had at our disposal on the high performance cluster at Sickkids Hospital. By way of comparison, registration for the Allen Brain Atlas for mice is often done at 20 micron isotropic. However, it should be noted that the downsampling of the images for registration is distinct from steps that would be used for cell counting/identification. We would expect users to acquire images at the resolution needed to detect individual cells and only downsample as needed during the registration process. The downsampling would have no effect on the ability to assign individual cells to specific anatomical structures given that the volume of the brain regions in the atlas are well above 8 µm^3^ (e.g, the smallest region is 26,496 µm^3^; for all volumes; Supplementary file 2).

3) How many different fish did they image? We now know there are a LOT of differences even between siblings of wild-type lines. This would provide another form of error they would need to address, if nothing else for users to explore their resource.

All fish were from the AB strain. For the TO-PRO stain, we averaged together images from 17 fish (line 532). For antibody stains we used images from at least five animals (lines 187-188). Because our goal was to generate a representative brain that did not reflect potential idiosyncrasies of individual animals, we averaged images from multiple fish together. However, the inclusion of individual data points in our analysis of sex and brain volume will allow readers to explore the variability of different brain region sizes across animals (Figure 5 —figure supplements 2 and 3).

Reviewer #2 (Recommendations for the authors):My comments below should be interpreted only as efforts to make this atlas as user-friendly as possible.Understandably, due to the incomplete knowledge of the zebrafish brain, some parts of the brain have not been segmented. At present, these areas are indexed as 0 (zero), which is the same as the area outside the brain. I would strongly encourage the authors to distinguish unannotated voxels from space outside of the brain, as this will be essential to facilitate computational analyses of brain imaging data.

This has now been addressed by the addition of “unknown” regions in the segmentation (see comment above under ‘essential revisions’ for more details).

I suspect that the annotation file "2021-05-02_AZBA_Label_descriptions.txt" is formatted for ITK-SNAP. However, it could be more useful for other computational studies:(1) At present, the file contains only region abbreviations, making it necessary to constantly consult Table S2 of the manuscript. I would like this file to additionally include the full name of the region.

Full names are now included in the file label descriptions file for ITK-SNAP that are part of the Dryad repository.

(2) Each region should be associated with a major brain division (telencephalon, diencephalon etc) and where possible, given any other useful annotations (ventricle, tract, nucleus). These labels provide valuable grist for computational methods.

We have now included ontological levels for all brain regions in the atlas as part of a more extensive supplemental data (Supplementary file 2) and included a structure tree (Supplementary file 3).

(3) For people not using ITK-SNAP it would be helpful to provide the coordinates of a voxel within each region. For example I tried in vain to hunt for the voxels corresponding to MAC.

We have now included this information with the additional supplemental table (Supplementary file 2) and have put together a website to display and explore the images (azba.wayne.edu). I have also put together a short video tutorial on using ITKSNAP to explore the atlas that includes another method for identifying individual brain regions (https://youtu.be/uVLqFJd4LDk) with this link also available on the website and the Dryad repository.

(4) I understand the first column (index) and last column (abbreviation). What are the other columns?

The other columns represent the Red, Green, and Blue components of the segmentation coloration and whether or not (0 or 1) the segmentations are visible when loaded into ITK-SNAP. The current version of the file (“2021-08-22_AZBA_label_descriptions.txt”) contains a header that describes the file format. We apologize for removing this header in the previous upload.

After light-sheet imaging, what actually happens during the step referred to as "3D volume generation", or "Three-dimensional volumes were created from individual image sets", or "Image stacks from individual fish brains were converted to 3D volumes" ? Does this simply refer to the stitching plus resampling procedure?

Yes, images were stitched and resampled to 4 micron isotropic and turned into.mnc files for use in registration.

If I understand correctly, the ultimate step in generating a template brain was to mirror the first average, then register the previously linearly aligned TOPRO samples to to it. Although I appreciate from the data I retrieved from Dryad that the result is a symmetric brain, I can't quite understand why this procedure yielded a symmetric template, rather than just a more precise template with left/right reversal, so please describe what happened in this step more clearly.More importantly, while in general a symmetric template is desirable (especially for computational approaches), this procedure obscures important biological asymmetries present in the habenula, a major focus of neurobiological studies in zebrafish. Could you somehow exclude the habenula from the symmetrization step?

Symmetry was only used for the target of the registration process, not in the generation of the final image. The use of a symmetric target was to make the final average of all registered images in line with the imaging planes. Thus, the average brain for the TOPRO signal is not tilted across axes but instead parallel to the axes and the images. This was used to create the initial TOPRO and autofluorescence signal average image. Once this average image is parallel with the image axes then this step is no longer used. No image in the final atlas space has been made symmetric. We have made edits to our text to clarify this by removing the word symmetric to describe the average brain from the second pipeline (lines 550-552).

No information is provided on the precision of the registration procedure. It would be helpful to assess precision so that subsequent users can evaluate whether their own procedures provide sufficiently close alignments.

We have now assessed the precision of our registration using both the TO-PRO signal and the autofluorescence signals (Figure 4 —figure supplement 1). Although we found the TO-PRO image to give greater precision (15 vs 99 µm), there are some caveats to this comparison. Namely, because the autofluorescence images lack cells, it is more difficult to identify common landmarks in the autofluorescence images than the TOPRO images. This is supported by the greater amount of variability in our registration precision for the autofluorescence versus TO-PRO images (Figure 4 —figure supplement 1). We have now included our findings and a discussion of this caveat in the manuscript (results: lines 193-209; methods: lines 565-570; Figure 4 —figure supplement 1).

The Methods are generally appropriately detailed, and specify the age of the fish. However, 3-4 month old fish can vary widely in size, and therefore presumably in brain volume. It may be helpful to add details on the size of the fish, and how sensitive the accuracy of the registration procedure is to size.

Unfortunately, we did not weigh the fish when the brains were collected to determine size. Nonetheless, we have now assessed the brain volumes of all the fish used in the TO-PRO registration (Figure 5 —figure supplement 1 and 2) which also provides an estimate of the variability in brain size in our samples (~2 to 3.5 mm^3^).

I expect that autofluorescence is more variable than TOPRO and carries less information. However there is no information provided for evaluating the quality of the registration achieved using autofluorescence as a bridging channel. Figure 4D (which is small and hard to see) purports to illustrate the accuracy of autofluorescence bridging registration using the example of the Locus Coeruleus. However, because one can not reliably ascertain the LC in either the HuC or TOPRO stain, there is nothing to usefully compare the TH stain to. Precise registration is important because the IHC channels used for segmentation were registered using autofluorescence. Please provide a measurement of registration accuracy using autofluorescence. Why was TOPRO not used with IHC for this step, to provide the strongest alignment?The Methods are generally appropriately detailed, and specify the age of the fish. However, 3-4 month old fish can vary widely in size, and therefore presumably in brain volume. It may be helpful to add details on the size of the fish, and how sensitive the accuracy of the registration procedure is to size.Much of the segmentation was performed on coronal sections, for which region boundaries are generally smooth. However, when viewed in other planes, boundaries are jagged resulting in artificial discontinuities within regions. It would be desirable to implement at least a simple procedure to smooth region boundaries in 3D space.

We very much appreciate the reviewer pointing this out as it pushed us to figure out a way to make segmentation smooth in all three planes. This has significantly improved the quality of the segmentation across the three axes.

Reviewer #3 (Recommendations for the authors):In the intro page 4 the authors claim the adult zebrafish brain would be "several orders of magnitude larger" than the larval brain. "Several" to me means more than two, but I would think that the adult zebrafish brain is just one order of magnitude larger (larva 500 micrometer, adult 5-15 mm).

We thank the reviewer for pointing this out. As the reviewer notes, the size of a larval brain is estimated to be ~0.3 mm^3^. We find our adult zebrafish to have a brain volume of approximately 3 mm^3^. The manuscript has been updated to reflect this (lines 9596).

Some images, e.g. in Figure 4, especially 4d, but also parts of Figure 7, are provided with too low resolution in the PDF to judge detail.

We agree that it is difficult to fully appreciate the full detail of the atlas from traditional manuscript figures. To facilitate exploration of the atlas in greater detail, we have now created a website that is broadly accessible and straightforward to navigate (azba.wayne.edu). We now mention the existence of this website in the abstract (line 45) and the body of the manuscript (line 327). Even greater detail can be explored using the free, user friendly, open-source software, ITK-SNAP. To facilitate the use of ITK-SNAP, we have created a YouTube video tutorial with how to load and explore the atlas after downloading the files from Dryad (https://youtu.be/uVLqFJd4LDk).

Figure 7B middle panels "0,10"Prox1 appears to be an example that a major expression domain maps to a non-annotated anatomical domain – or the prethalamus is so dark here that it cannot be distinguished from black. In the text where Prox1 is presented, this domain appears to be not mentioned.For some panels it may be better to present cutouts at higher magnification, with anatomical regions outlined in fine white lines and labelled (rather than using the color code only).

We have now labelled all non-annotated domains as ‘unknown’ (lines 227-234). Instead of presenting cutouts, as suggested, we now point the reader to our website (azba.wayne.edu) to examine the atlas in greater detail.

Figure 7D GFAP: I cannot identify in the figure the GFAP pattern described in the text:"we found GFAP most concentrated near the midline and ventricles (Figure 7D)."Why is there no GFAP signal in the midsagital section 7D 0.00?

We respectfully disagree with the reviewer. Of the three sagittal images shown for GFAP (Figure 7D), the midsagittal section (0.00) shows the most staining. This may be better appreciated by browsing the atlas online (azba.wayne.edu).

Materials:Given sex differences in the fish brain (see papers cited by authors), the authors should report whether male or female brains or a mix of brains from both sexes were used.

We used a mix of male and female fish for imaging (~50% each). We now more explicitly state this in the Results section (line 151).

LimitationsGiven that relatively few antibodies are available for specific zebrafish neuronal markers, it would be very informative to know if the technique used by the authors would in principle be applicable to fluorescent whole mount hybridization stained brains.

In the manuscript we cite previous work has successfully combined in situ hybridization chain reaction and iDISCO (Kramer et al., 2018; line 100).